# Commercial Aircraft-Assisted Suicide Accident Investigations Re-Visited—Agreeing to Disagree?

**Alpo Vuorio** [1,2,*] 🆔**, Robert Bor** [3,4]**, Antti Sajantila** [2,5] 🆔**, Anna-Stina Suhonen-Malm** [1] **and Bruce Budowle** [2]

1   Mehiläinen Airport Health Centre, 01530 Vantaa, Finland
2   Department of Forensic Medicine, University of Helsinki, 00014 Helsinki, Finland
3   Royal Free Hospital, Pond Street, London NW3 2QG, UK
4   Centre for Aviation Psychology, London NW3 1ND, UK
5   Forensic Medicine Unit, Finnish Institute for Health and Welfare, 00271 Helsinki, Finland
*   Correspondence: alpo.vuorio@gmail.com

**Abstract:** Background: The number of aircraft-assisted suicides can only be considered a rough estimate because it is difficult and, at times, impossible to identify all cases of suicide. Methods: Four recent reports of accidents occurring in 1997 in Indonesia, 1999 in Massachusetts in the United States, 2013 in Namibia, and 2015 in France related to commercial aircraft-assisted suicides were analyzed. This analysis relied on data extracted from the accident reports that supported aircraft-assisted suicide from the: (a) cockpit voice recorder (CVR) and flight data recorder (FDR), (b) medical history, (c) psychosocial history, (d) toxicology, (e) autopsy, and (f) any methodology that utilized aviation medicine. There are some limitations in this study. Although all analyzed accident investigations followed ICAO Annex 13 guidelines, there is variability in their accident investigations and reporting. In addition, accident investigation reports represent accidents from 1997 to 2015, and during this time, there has been a change in the way accidents are reported. The nature of this analysis is explorative. The aim was to identify how the various aircraft accident investigators concluded that the accidents were due to suicidal acts. Results: In all four accident reports, FDR data were available. CVR data were also available, except for one accident where CVR data were only partially available. Comprehensive medical and psychosocial histories were available in only one of four of the accident reports. Conclusion: To prevent accidents involving commercial aircraft, it is necessary to identify the causes of these accidents to be able to provide meaningful safety recommendations. A detailed psychological autopsy of pilots can and likely will assist in investigations, as well as generate recommendations that will substantially contribute to mitigating accidents due to pilot suicide. Airborne image recording may be a useful tool to provide additional information about events leading up to a crash and thus assist in accident investigations.

**Keywords:** aircraft-assisted suicide; accident investigation; CVR; FDR; airborne image recording; psychological autopsy





## 1. Introduction

Research into aircraft-assisted pilot suicides is based on official accident investigation reports that analyze and conclude the likely cause of an accident [1,2]. Prior to the Germanwings aircraft-assisted suicide in 2015, only a few research articles had analyzed aircraft-assisted suicides, and these analyses focused on general rather than commercial aviation [3–6]. In general aviation, there is generally no cockpit voice recorder (CVR) or flight data recorder (FDR) data. In these reports, analysis of aircraft accident suicides in general aviation is based on autopsy and pilot medical history, which in many cases is incomplete and brief, and may contribute to inconclusive or best assumptions from the evidence of suicide findings.

According to the ICAO Annex 13 [7], "The State conducting the investigation shall send a copy of the draft Final Report to the following States inviting their significant and

substantiated comments on the report as soon as possible: (a) the State that instituted the investigation; (b) the State of Registry; (c) the State of the Operator; (d) the State of Design; (e) the State of Manufacture; and (f) any State that participated in the investigation as per Chapter 5". Based on accident reports in the United States over almost a 20-year period (1993–2021) 24/7244, fatal accident cases in general aviation were concluded to be due to aircraft-assisted suicides (0.33% 95% CI 0.21–0.49) [1]. This estimate, and others, of aircraft-assisted suicides, can only be considered as a rough estimate because it is difficult and, at times, impossible to identify all cases of suicide. Furthermore, many cases involve private (single-acting) pilots or those of very small operators where investigations may be more limited, as shown in the analysis of accidents related to aircraft-assisted suicides in the USA [1], as opposed to the rarer intentional crashes occurring with large passenger aircraft. Several factors need to be analyzed before judging an accident to be caused by pilot suicide.

It is very important to define what is meant by 'suicidal behavior'. Goodfellow et al. [8] conducted a systematic literature review and analyzed 29 articles and book chapters. They concluded that there are four main factors related to assessing suicidal behavior: agency, knowledge of fatal outcome, intent, and outcome, which will aid in understanding the meaning of suicide in different cultures. It is therefore important to note that the nomenclature of suicidal behaviors varies and makes comparisons between different challenging studies [9].

In a comprehensive comparative study by Mann et al. [10] of 14–72-year-old patients who had attempted suicide following admission to a psychiatric clinic with patients who had never attempted suicide, the severity of their current depression did not distinguish these two groups. Therefore, the severity of mood was not found to be a predictor of suicidal behavior. Suicide-related factors were also studied recently in a systematic review [11]. For example, some of these factors related to suicide are any mental disorder, personality disorder, history of self-harm, relationship conflict, and high scores on brief assessments or screening tests for suicidal ideation. Mann et al. [10] found several comorbidities related to suicidal behavior. These comorbidities included past substance use disorder or alcoholism and a family history of suicidal acts.

A study of road accidents in Sweden (i.e., a re-analysis of a 2012 study) by Andersson and Sokolowski [12] showed that through extended psychosocial investigation, the number of accidents due to suicide was 63% higher than originally estimated. Accident investigation related to traffic accidents needs to include the socio-economic background as well as psychosocial and mental health factors. The factors that need to be collected are very similar to those included in the Columbia-Suicide Rating Scale [13,14]. Collecting socio-economic background can be comparable to that of a psychological autopsy (PA) which is a forensic assessment to evaluate the deceased's mental 'condition' (as 'state' cannot be assessed retrospectively unless the pilot was seen by a medical practitioner shortly before the crash) and likely intent. This may include rehearsals for the act, such as through flight simulator scenarios being tested and the social and financial circumstances of the pilot being investigated to see whether these may be linked to personal life challenges and emotional distress. Also, the impact of the COVID-19 pandemic needs to be considered [15].

Given the uncertainty and different factors that can be applied, we analyzed four accident reports which have been related to commercial aircraft-assisted suicides that occurred between the years 1997 and 2016 [16–19]. The aim was to identify how the various aircraft accident investigators concluded that the accidents were due to suicidal acts. Based on these analyses and experience, recommendations to mitigate future aircraft-assisted suicides are considered to investigate accidents in which there are suspicions of possible aircraft-assisted suicide [20].

Additionally, we mention in the discussion two accidents that potentially could be due to aircraft-assisted suicides. The first is the Malaysian Airlines flight MH370 which on 8 March 2014 disappeared soon after a routine handover to Vietnam air traffic control. Despite an extensive search, the aircraft has not been located [21]. The second is the China

Eastern Airlines flight 5735, which on 21 March 2022, suddenly descended steeply in mid-flight and struck the ground at high speed [22]. This aircraft accident investigation is ongoing.

## 2. Materials and Methods

Four commercial aircraft accidents were included in this study. The first analyzed accident occurred in Indonesia on 19 December 1997 [16]. SilkAir flight MI 185 was an operating flight traveling from Jakarta Soekarno-Hatta International Airport to Singapore Changi Airport. The airplane descended from its cruising altitude of 35,000 feet and impacted the ground. The FDR stopped recording several times before the descent, and there were no mayday calls from the pilots.

The second analyzed accident occurred on 31 October 1999, when EgyptAir flight 990 crashed into the Atlantic Ocean South of Massachusetts [17]. About 29 min after take-off, the FDR showed that the first officer disconnected the autopilot. The FDR recorded an abrupt nose-down elevator movement, and the aircraft began to descend. The CVR recorded that the first officer said several times, "I rely on God". After that, the FDR recorded additional, slightly larger inboard aileron movements. A master warning alarm began to sound, and the captain, who was returning from the toilet, was wondering what was happening. The aircraft crashed about two minutes later.

The third analyzed accident occurred on 29 November 2013 with Mozambique Airlines flight 470 [18]. After 1 h 50 min into the flight, the First Officer stated he had to go to the toilet. Then the captain handled the auto flight system leading to a sustained descent and collision with the terrain. Just before the collision, sounds were heard of someone trying to open the cockpit door.

The fourth analyzed accident occurred on 24 March 2015 with Germanwings flight 4U9525 [19]. In the cruise phase of the flight, the first officer waited until he was alone in the cockpit. Then he modified the autopilot settings to order the airplane to descend and kept the cockpit door locked. The first officer did not respond to the calls from air traffic controllers, and the aircraft impacted the terrain in the French Alps.

The above-mentioned four accident reports related to commercial aircraft-assisted suicides were analyzed [16–19] (Table 1). The nature of this analysis is explorative. While the number of aircraft-assisted suicides is extremely low in commercial aviation, the consequences are catastrophic and resonate worldwide. Therefore, it is important to assess the events to gain insight into possible preventative measures that can be enacted. We chose four recent accidents to demonstrate what the accident investigators' reasonings are based on how they concluded that they were attributable to pilot suicide. In addition, in the two accidents selected for analysis, divergent opinions emerged in the commentaries of the research reports. Although limited in sampling, these differences made these reports particularly informative.

We used content analysis, and one of the authors (AV) analyzed in detail the accident reports, and other authors commented on the analysis. This analysis relied on data extracted from the accident reports that supported aircraft-assisted suicide from (a) CVR and FDR, (b) medical history, (c) psychosocial history, (d) toxicology, (e) autopsy, and (f) any methodology that utilized aviation medicine. Because the state of registry, the state of the operator, the state of design, and the state of manufacture shall each be entitled to appoint an accredited representative to participate in the investigation, we take into account, additionally, the critical responses by these shareholders in two of the cases [16,17]. Also, safety recommendations to mitigate future similar kinds of accidents were analyzed regarding the two more recent accidents [18,19].



**Table 1.** Analyzed accidents [10–13].

| Accident, Country, Date, Flight | Aircraft | Accident Suspected Pilot Related to Accident Pilot. Age (Years) | Flight (Hours) Last Medical Examination (Category) | Fatalities Crew Passengers | Accident Investigation Authority |
|---|---|---|---|---|---|
| Indonesia, 19 December 1997, MI185 | Boeing 737–200 | Indonesia Captain, 41 | 7173, 1 November 1997 (ATPL) | 7 97 | National Transportation Safety Committee (NTSC), Indonesia |
| Massachusetts, USA, 31 October 1999, MSR990 | Boeing 767–366ER | Massachusetts, USA First officer, 59 | 12 538, 19 June 1999 (ATPL) | 14 203 | National Transportation Safety Board (NTSB), Washington, USA |
| Namibia, 29 November 2013, TM479 | C9-EMC | Namibia Captain, 49 | 9052, August 2013 (ATPL) | 6 27 | Directorate of Aircraft Accident Investigation, Namibia |
| France, 24 March 2015, 4U9525 | Airbus 320–211 | France First officer, 27 | 919, 11 February 2014 [MPL(A)] | 6 144 | Bureau d'Enquêtes et d'Analyses pour la sécurité de l'aviation civile (BEA), France |

## 3. Results

These accidents were investigated by the national aircraft accident investigation authorities from Indonesia, the United States, Namibia, and France, respectively [16–19]. In total, 33 cabin crew members and 471 passengers died due to these accidents. In two cases, the suspected pilot related to the accident was the first officer, and in two cases, it was the captain. All the suspected pilots were males and between the ages of 27 and 59 years (Table 1).

In all four accident reports, FDR data were available [16–19]. CVR data were also available except for the Indonesia flight MI185 (1997), where CVR data were only partially available [10].

Medical and psychosocial histories were available in only one [19] of four accident reports (Table 2). In two accidents, psychosocial histories were partly available [16,18].

**Table 2.** Documented medical and pathological information collected in accident investigations [16–19].

| Analyzed Factor in Accident Investigation Report | Indonesia, 19 December 1997, MI185 | Massachusetts, USA, 31 October 1999, MSR990 | Namibia, 29 November 2013, TM479 | France, 24 March 2015, 4U9525 |
|---|---|---|---|---|
| Medical history (years) | - * | - ** | - *** | 8 years |
| Medication(s) based on clinical data | - | - | - | + |
| Doctor's/hospital visits | - | - | - | + |
| Psychology visits | - | - | - | + |
| History of medical certifications | 1 year | 1 year | 1 year | 8 years |
| Doctor involved in analysis | - | - | - | + |
| Special doctor's team involved in analysis providing postmortem diagnosis | - | - | - | + |
| Postmortem toxicology | - | - | + | + |
| Identification | - | - | + | + |

\* Statement that no significant medical history was related to the crew. \*\* Close friend interviewed. \*\*\* Information regarding medical certification and some social information under the title, "Human factors".

In two accidents, the accident investigation authorities that conducted the investigations and the aviation authority that commented on the investigation had different interpretations as to the cause of the accident (Table 3) [23,24]. According to ICAO Annex 13 protocol, the State of Registry and the State of Manufacturer also can comment on the investigation.

**Table 3.** Accidents Occurred in Indonesia and Massachusetts: Official Accident Report vs. Response to Accident Investigation Draft Report [16,17].

| Accident, Country, Date, Flight | Accident Investigation Report Safety Investigation Authority, Summary and Probable Cause | Response to Accident Investigation Draft Report |
|---|---|---|
| Indonesia, 19 December 1997, MI185 | National Transportation Safety Committee There was no evidence indicating that the performance of either pilot was adversely affected by any medical or physiological condition. There was no evidence that there were difficulties between pilots. Until the stoppage of the CVR, pilots conducted flight normally. There was no evidence of a technical cause of the accident. The investigation yielded very limited data to make conclusions possible. | NTSB There was no technical failure explaining the accident. The accident can be explained by intentional pilot action. The flight profile was consistent with sustained manual nose-down flight control inputs, most probably made by the captain. CVR was intentionally disconnected. Recovery was possible but not attempted. |
| Massachusetts, USA, 31 October 1999, MSR990 | NTSB The nose-down movements did not result from a technical failure. The initial airplane's movements were due to manipulation by the first officer. After the captain returned to the cockpit, he tried to recover the aircraft while the first officer pushed the nose down. The probable cause for the accident was the first officer's flight control inputs. | Egyptian Aircraft Accident Investigation Directorate NTSB has not done the necessary type of professional accident investigation. The report selectively uses the facts of the investigation to develop, support and "prove" the findings of the deliberate act. The report fails to account for specific facts in the record that refute the possibility of an intentional act. |

NTSB = National Transportation Safety Board.

Regarding the accident in Indonesia in 1997, the local National Transportation Safety Committee did not find the accident intentional, while the NTSB, in contrast, considered the accident intentional [16]. The NTSB report regarding the accident in Massachusetts considered it intentional, while the Egyptian Aircraft Accident Investigation Directorate held the opposite view [17]. Unfortunately, there were only a limited amount of medical investigation data regarding both accidents.

The safety recommendations because of the accident in Namibia were primarily oriented toward solutions related to the technical operation and the tracking of the aircraft [18] (Table 4). This focus diminished the centrality of the pilot murder–suicide. The safety recommendations because of the accident in France were related to the medical monitoring of pilots and the expansion of emotional support for pilots [19].

**Table 4.** Safety recommendations related to the accidents in Namibia [18] and France [19].

| Safety Recommendations Namibia, 29 November 2013, TM479 * | Safety Recommendations France, 24 March 2015, 4U9525 ** |
|---|---|
| Mozambique CAA should ensure the procedure of two people in the flight deck | EASA requires that when a class 1 medical certificate is issued to an applicant with a history of psychological/psychiatric trouble of any sort, conditions for the follow-up of their fitness to fly be defined. |
| ICAO should establish a working group that creates threat management emanating from both sides of the cockpit door | EASA includes in the European Plan for Aviation Safety an action for the EU Member States to perform routine analysis of in-flight incapacitation, with reference but not limited to psychological or psychiatric issues. |
| ICAO should establish a working group to review installation visual recordings inside and outside the cockpit | EASA, in coordination with the Network of Analysts, performs routine analysis of in-flight incapacitation with reference to psychological or psychiatric issues. |
| Early warning system regarding abnormal flight behavior needs to expedite by ICAO | EASA ensure that European operators include in their management Systems measures to mitigate socio-economic risks related to a loss of license. |
| Aircraft tracking and localization system other than ELT needs to expedite by ICAO | EASA defines the modalities under which EU regulations would allow pilots to be declared fit to fly while taking anti-depressant medication under medical supervision. |
| | The World Health Organization develop guidelines for its Member States to help them define clear rules to require healthcare providers to inform the appropriate authorities when a specific patient's health is likely to impact public safety. |
| | EASA ensure that European operators promote the implementation of peer support groups to provide a process for pilots, their families, and peers to report and discuss personal and mental health issues, with the assurance that information will be kept in confidence in a just-culture work environment, and that pilots will be supported as well as guided to provide them with help. |

* Quoted: [18]; ** quoted: [19].

## 4. Discussion

Although suicide by deliberately crashing an aircraft is a rare phenomenon, its significance for flight safety and passengers' perception of aviation safety is important. This phenomenon is, to some extent, related to significant disasters occurring in society, and it is still too early to assess whether the COVID-19 pandemic has impacted suicidal behavior currently or will in the near future [15,25–27]. In investigating air accidents, identifying and/or verifying is due to assisted pilot suicide presents very special challenges [1].

The analysis of these four accident investigation reports revealed that the FDR and CVR information lends significant weight in determining whether the accident resulted from the pilot's aircraft-assisted suicide. FDR registration was used in all four analyzed accident investigation reports. In addition, CVR registration was used in three cases, and only partial CVR registration was used in one case [16–19]. The conclusion of a suicidal accident arose from the fact that no technical fault was found with the plane, and the trajectory of the plane was exceptional. For example, an accident under investigation is flight MU5735 which occurred on 21 March 2021. According to the media, no mechanical failure was found with the aircraft; the FDR analysis preliminarily supports a potential intentional aircraft crash [22]. The final accident report has yet to be disclosed. The CVR registration was used to support that the abnormal flight path was caused by the pilot's intentional action. In the accidents that occurred in Namibia in 2013 and in France in 2015, the CVR registration supported that the pilot's deliberate, controlled actions led to the accident [18,19].

The remaining commercial aircraft crashes of significance occurred in Indonesia in 1997 and Massachusetts, USA, in 1999 [16,17]. Interpretation of the information of CVR data was not entirely clear, which may have contributed to disagreement between accident investigation authorities [16,17]. The Directorate of Aircraft Accident Investigation of Namibia recommended that the ICAO should establish a working group to review the installation of visual recordings inside and outside the cockpit [28]. The issue of airborne image recorders and Annex 6 defining this approach has been discussed for several years in

the ICAO [29]. The ICAO's working paper mentions several commercial airplane accidents where airborne image recorders would have provided additional information to assist in the accident investigation [29]. In general, airborne image recorders could be useful in cases of unlawful interference related to flight operations. Currently, there is an attempt by the ICAO to mandate airborne image recorders [28].

A psychological autopsy (PA) is a forensic assessment that evaluates the deceased's mental state and intent [30]. The purpose of a PA is to (a) obtain the motive and cause of death, (b) analyze conditions surrounding the death, (c) analyze factors that impacted suicide risk, (d) help family members and other shareholders with their restorative mechanisms, and (e) help in administrative functions [31]. It also helps in these circumstances to determine the manner of death (natural, accidental, suicide, homicide). Due to the complexities related to suicide, this method has limitations, but it can be useful in addition to individual potential suicide investigations and research [30,32,33]. Isometsä [34] concluded in his review that PA is "one of the most valuable tools of research on completed suicide". It can be concluded that it synthesizes information from multiple resources. It can be applied to suicides that have occurred in other occupations, such as in the building and healthcare industries [35,36].

PA is useful to use in aviation medicine-related mental health cases. This evaluation needs to be used in cooperation with all those specialists involved in an accident investigation: accident investigators will likely call on the expertise of aviation and forensic psychologists, psychiatrists, the pilot's aeromedical examiner(s) (AME), regular personal doctors and human factors and flight safety personnel. Despite this evaluation, there is rarely absolute certainty about suicidal intent being the sole cause of an accident. In many accident investigation cases, the main effort has been to analyze health information obtained from AME through periodic medical examinations. However, several studies have shown that pilots often only share some of their health information with the aviation doctor [37,38]. It is also possible that the seriousness of the pilot's psychiatric illness has not been recognized. Early diagnosis of major psychiatric illnesses, like psychosis, may be challenging even for psychiatrists [39]. But recognizing psychotic symptoms is essential; a recent study has shown that psychotic symptoms associated with severe depression doubled the risk of suicide compared to the risk of severe depression without psychotic symptoms [40].

Unlike more specific and arguably more clear-cut physiological determinants or the course of events, causes, and consequences of an air crash, such as cardiac or neurological symptoms, mental health, and well-being factors, are less straightforward to determine. Their role may be a cofactor alongside others to assess whether human factors or mental ill health may be factors in an aircraft crash. Investigations will typically focus on several factors, which include (a) past and recent mental health and well-being history, (b) training records and any challenges or difficulties noted at the start of the pilot's career, (c) past and present family history of mental ill health, (d) relationship status, stability, dating patterns, and challenges, (e) financial circumstances and difficulties, (f) employment and performance records with a focus on resilience and challenges, (g) occupational stress and security, (h) substance use and misuse history, and (i) general life 'behaviors', such as internet use, addictions and dependencies, legal challenges and problems, and medical issues and problems. Personality and past behavioral issues and patterns are likely reasonable predictors of future behavior. They will be subjects in an investigation, as they can help shed light on the current accident and the pilot's behavior. An example of a specific type of aircraft accident is flight MH370, which occurred on 8 March 2014 [21]. The report is 450 pages long, and under the title "Significant past medical and medication history", there are only about 20 lines of text. It is possible there were no health issues with the pilots to report; however, the aircraft has not been found, and a carefully conducted psychological autopsy was not undertaken. In the current report, it is difficult to follow up on how the medical history was covered.

While this article focuses particularly on the significance of the psychological autopsy, it should be noted that aviation pathology and toxicology can contribute substantially to clarifying the accident [41–44]. Among these four analyzed accidents, two also included toxicological studies [18,19]. In the accident in France, both escitalopram and mirtazapine were found in the analysis [13]. Interestingly, there is a potential interaction between these drugs, as consuming escitalopram with mirtazapine may increase the risk of a rare but serious condition known as serotonin syndrome [45]. The prevalence of this symptom is very low (in a large cohort study, less than 0.2% of patients received serotonergic agents), and the symptom varies from mild to severe [46,47]. Also, interactions of psychiatric drugs, as well as between each other or with other drugs, may cause behavioral changes which need to be taken into consideration [48].

Altogether, different psychiatric drugs, as well as interactions of these drugs, may cause behavioral changes which need to be taken into consideration. Another issue to consider are the varying distributions of a drug(s) between body fluids and tissues [49]. Unfortunately, trying to relate the postmortem drug concentrations to those drug concentrations when the person was still alive is very demanding and, at times, impossible.

The main purpose of an accident investigation is to prevent similar accidents from occurring again. To this end, accident investigation reports provide safety recommendations. In this analysis, two accident reports provided mitigation recommendations to prevent aircraft-assisted suicides [18,19]. The safety recommendations provided after the accident in Namibia were more technical in nature, and operations related compared to those given after the accident in France by the Bureau d'Enquêtes et d'Analyses pour la sécurité de l'aviation civile (BEA) were focused on psychiatric aviation health, support, and duty of notification [18,19]. We proffer that the recommendations of this French accident investigation report have contributed to the advancement of aviation safety. As an example of this positive impact, there is EASA's recent report on Mental Health for Aviation SAFEty (MESAFE) [50]. This report covers, among other issues, mental health assessment methods.

## 5. Conclusions

In summary, to prevent what may arguably be preventable pilot suicide accidents involving commercial aircraft, it is necessary to identify the causes of these accidents to be able to provide meaningful safety recommendations. As shown in the accident investigation of Germanwings flight 4U9525, a detailed psychological autopsy on pilots can and likely will assist in investigations, as well as generate recommendations that will substantially contribute to mitigating accidents due to pilot suicide. Airborne image recording may provide additional information about events before the crash, which may help assist in an accident investigation. Use of this registration should be expedited. This recording would have given additional valuable information, especially in the accidents of Egypt Air flight 990 and Silk Air flight 185. Unfortunately, in an aviation medicine clinical setting, over-reliance on the known risk factors or the use of scales may create false reassurance, though comprehensive psychosocial assessments when there are risk factors could be useful [51]. There also is a need to research the detailed neuropsychiatric history targeted to those pilots who have had suicidal ideation [20].

## 6. Limitations

There are some limitations in this study. Although all analyzed accident investigations followed ICAO Annex 13 guidelines, there is variability in their accident investigations and reporting. This variation makes the comparisons somewhat difficult. In addition, accident investigation reports have evolved from 1997 to 2015. Currently, the purpose is not to find individual causes of the accident. Instead, the investigation report includes a description of the accident, the factors that led to the accident, and the consequences of the accident, as well as safety recommendations addressed to the relevant authorities and other actors for such measures as are necessary to increase public safety, prevent new accidents and

dangerous situations, combat damage, and enhance the operations of rescue and other authorities.

**Author Contributions:** A.V.: writing the first draft. A.V., R.B., A.-S.S.-M., A.S. and B.B.: editing to produce the final draft. All authors have read and agreed to the published version of the manuscript.

**Funding:** This research received no external funding.

**Institutional Review Board Statement:** Not applicable.

**Informed Consent Statement:** Not applicable.

**Data Availability Statement:** Not applicable.

**Conflicts of Interest:** The authors declare no conflict of interest.

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
