# Peer review of "Commercial Aircraft-Assisted Suicide Accident Investigations Re-Visited—Agreeing to Disagree?"

_safety_

Round 1
Reviewer 1 Report
A PDF file has been attached.

Author Response
|
Reviewer
|
|
|
I have reviewed the article “Commercial Aircraft Assisted Suicide Accident Investigations Re- visited –Agreeing to disagree?”. Even if the topic is one of a maximum value and impact for the aviation community and your endeavor is very much appreciated, the conclusions of your analysis are either too general (“it is necessary to identify the causes of these accidents to be able to provide meaningful safety recommendations” – is it not the major goal of any air crash investigation committee to identify the causes of the accident?) or they are already included in most of the air crash investigations (“A detailed psychological autopsy on pilots can and likely will assist in investigations” – this is quite often, if not always, performed by the air crash investigation committees).
|
Our response We have modified the sentence Unlike more specific and arguably more clearcut physiological determinants or the course of events, causes and consequences of an air crash, such as cardiac or neurological symptoms, mental health and well-being factors, are less straightforward to determine. |
|
In what concerns your third conclusion – “Airborne image recording may provide additional information about events before the crash which may be helpful in assisting in an accident investigation.” – you already mentioned at page 6, the last but one paragraph, that “Currently, there is an attempt by the ICAO to mandate airborne image recorders”. So, this conclusion also brings nothing new as there is already an ongoing procedure to mandate the usage of these recorders.
|
Our response
We have added text
Use of this registration should be expedited. This recording would have given additional valuable information especially in the accidents of Egypt Air flight 990 and Silk Air flight 185. |
|
I think that your decision to study this topic in general and these four reports in particular, was an inspired one, but perhaps a new, in depth analysis will help you reach more original conclusions that can be better tailored to the need of spotting out the pilots with suicidal behavior. Particular patterns that may be useful to track down in advance the pilots at risk to commit suicide, could perhaps, be identified within these reports.
|
Our response To the best of our knowledge this kind of analysis has not been carried previously and consider it to be an in depth analysis, although the number of cases is limited.
|
|
Under the present form, I am afraid that I cannot endorse the publication of your article.
|
Our response We have improved the study with the aid of two other referees and hopefully, the reviewers will find that it has been improved.
|
Reviewer 2 Report
Title of paper: Commercial aircraft assisted suicide accident investigations re-visited – Agreeing to disagree?
Manuscript Number: safety-2033056
Comments
The topic is very timely, and the review is within the scope of this journal. Commercial aviation is a complex social-technical system, and maintaining the safety of this transportation system is crucial. At the same time, we observe challenges put on operators in the system. Therefore, it is an excellent call to try and map features that might help the industry assess and act in exceptional cases to avoid similar situations in this paper.
Beyond that, the authors must address several concerns before this paper can be considered for publication. In the main, it demonstrates good thinking but a slightly relaxed approach to language and referencing, concern for the readers' understanding of the method or terminology and rooting in good academic writing practices. Please consider specific comments below and comments in the PDF of the initial submission.
Abstract
1. Consider adding details on methods used to reach conclusions as well as potential limitations to the presentation in the paper
Introduction
2. With due respect to the authors, a reader would benefit from stating the experience authors are referring to when stating this as an argument together with a relatively limited number of analyses to close this section.
Materials and Methods
3. Why would four accidents be enough to make conclusions? It would be good to offer some more reasoning about this.
4. It would be helpful to offer a critical response or view on the other two accidents, in addition to [10], [11]; even from broader literature sources (not necessarily government, e.g. ISASI or similar).
Discussion
5. A sudden introduction of COVID-19 in the discussion. It would be good if there is a preliminary reference or mention earlier in the paper to be used here in the discussion.
6. Well done on creating a checklist for Psychological autopsy (PA). Even better if you have clearly outlined the same and dedicated more explicit discussion/ presentation to the same.
7. Why not introduce MH 370 in the beginning with the other accidents and then use it in the discussion?
8. It would be good if there were some examples from other industries or cases where the psychological autopsy has already been used in other industries to indicate how useful it would be to introduce it into reviews of accidents in aviation.
9. Consider adding more details about the serotonin syndrome for a more generalist reader of the Safety journal in addition to the reference already offered.
10. Comments at the end of the section state the utility of the findings from one of the accidents. Is it better if the authors offered more confirmation of this from the literature? As it stands now, it is unclear whether it's the conclusion based on the feeling or arguments.
Conclusion
11. The conclusion states that imaging from the cockpit may help the investigation. It would be good to see from the literature earlier in the text if there is a case that this happens in some other industries. This practice might come from control rooms or laboratories, where these images help backtrack or find specific causes of incidents.
General Comments
12. Apply a more formal and less conversational writing style throughout the paper (e.g. of course; ). Please check the attached PDF for further examples of this.
13. We have observed a good selection of references, although dated at times. It would be better if you used some more Safety Journal references, if available. Consider adding.

Author Response
|
REVIEWER Title of paper: Commercial aircraft assisted suicide accident investigations re-visited – Agreeing to disagree? Manuscript Number: safety-2033056 Comments The topic is very timely, and the review is within the scope of this journal. Commercial aviation is a complex social-technical system, and maintaining the safety of this transportation system is crucial. At the same time, we observe challenges put on operators in the system. Therefore, it is an excellent call to try and map features that might help the industry assess and act in exceptional cases to avoid similar situations in this paper. Beyond that, the authors must address several concerns before this paper can be considered for publication. In the main, it demonstrates good thinking but a slightly relaxed approach to language and referencing, concern for the readers’ understanding of the method or terminology and rooting in good academic writing practices. Please consider specific comments below and comments in the PDF of the initial submission.
|
We thank you and provide point by point comments. |
|
Abstract Consider adding details on methods used to reach conclusions as well as potential limitations to the presentation in the paper We have added details of methods as well as relevant limitations.
We have also added an additional paragraph to the end of the discussion regarding the limitations.
|
Background: The number of aircraft assisted suicides can only be considered as a rough estimate, because it is difficult and at times not possible to identify all cases of suicide. Methods: Four recent reports of accidents, occurring in 1997 in Indonesia, 1999 in Massachusetts in the United States, 2013 in Namibia and 2015 in France, which have been related to commercial aircraft assisted suicides were analyzed. This analysis relied on data extracted from the accident reports that supported aircraft assisted suicide from the: (a) cockpit voice recorder (CVR) and flight data recorder (FDR), (b) medical history, (c) psychosocial history, (d) toxicology, € autopsy and (f) any methodology that utilized aviation medicine. There are some limitations in this study. Although all analyzed accident investigations followed ICAO Annex 13 guidelines, there is variability in their accident investigations and reporting. This variation makes the comparisons somewhat difficult. In addition, accident investigation reports represent accidents from the years 1997 to 2015, and during this time there has been a change in the way accidents are reported. The nature of this analysis is explorative. The aim was to identify how the various aircraft accident investigators arrived at the conclusion that the accidents were due to suicidal acts. Results: In all four accident reports flight data recorder (FDR) data were available. Cockpit voice recorder (CVR) data were available as well except for one accident where CVR data were only partially available. Comprehensive medical and psychosocial histories were available in only one of four of the accident reports. Conclusion: To prevent accidents involving commercial aircraft, it is necessary to identify the causes of these accidents to be able to provide meaningful safety recommendations. A detailed psychological autopsy on pilots can and likely will assist in investigations, as well as generate recommendations that will substantially contribute to mitigate accidents due to pilot suicide. Airborne image recording may be a useful tool to provide additional information about events leading up to a crash and thus assist in accident investigations. There are some limitations in this study. Although all analyzed accident investigations followed ICAO Annex 13 guidelines, there is variability in their accident investigations and reporting. This variation makes the comparisons somewhat difficult. In addition, accident investigation reports have evolved during the years 1997 to 2015. Currently the purpose is not to find individual causes of the accident. Instead, the investigation report includes a description of the accident, the factors that led to the accident, and the consequences of the accident, as well as safety recommendations addressed to the relevant authorities and other actors for such measures as are necessary to increase public safety, prevent new accidents and dangerous situations, combat damage, and enhance the operations of rescue and other authorities.
|
|
Introduction 2. With due respect to the authors, a reader would benefit from stating the experience authors are referring to when stating this as an argument together with a relatively limited number of analyses to close this section.
|
Our response: We have added the following text and references — see the red text. Prior to the Germanwings aircraft assisted suicide in 2015, there have been only a few research articles that have analyzed aircraft assisted suicides and these analyses focused on general rather than on commercial aviation (Bills et al., 2005; Lewis et al., 2007; Schwark et al., 2008; Cullen et al., 2007). In general aviation, there is generally no cockpit voice recorder (CVR) and flight data recorder (FDR) data. In these reports, analysis of aircraft accident suicides in general aviation is based on autopsy and pilot medical history which in many cases is incomplete and brief, which may contribute to inconclusive or best assumption from the evidence of findings of suicide. References Bills CB, Crabowski JG, Li G. Suicide by aircraft: a comparative analysis. Aviat Space Environ Med 2005; 76:715–9. LewisRJ, JohnsonRD, WhinneryJE, ForsterEM. Aircraft-assisted pilot suicides in the United States, 1993-2002. Arch Suicide Res 2007; 11:149–61 Schwark T, Severin K, Grellner W. “I am flying to the stars”— suicide by aircraft in Germany. Forensic Sci Int 2008; 179:e75–8. Cullen SA, Drysdale HC, Mayes RW. Role of medical factors in 1000 fatal accidents: case note study. BMJ 1997; 314:1592. |
|
Materials and Methods 3. Why would four accidents be enough to make conclusions? It would be good to offer some more reasoning about this.
|
Our response See kindly the first paragraph of methods. We have added the red text to clarify methodology. The above mentioned four accident reports which have been related to commercial aircraft assisted suicides were analyzed [10-13] (Table 2). The nature of this analysis is explorative. While the number of aircraft assisted suicides is extremely low in commercial aviation, the consequences are catastrophic and resonate throughout the world. Therefore, it is important to assess the events to gain insight into possible preventative measures that can be enacted. We chose four recent accidents to demonstrate what the accident investigators’ reasonings are based on how they concluded that they were attributable to pilot suicide. In addition, in the two accidents selected for analysis, divergent opinions emerged in the commentaries of the research reports. Although limited in sampling, these differences made these reports particularly informative. We used content analysis and one of the authors (AV) analyzed in detail the accident reports and other authors commented the analysis. This analysis relied on data extracted from the accident reports that supported aircraft assisted suicide from: (a) CVR and FDR, (b) medical history, (c) psychosocial history, (d) toxicology, € autopsy and (f) any methodology that utilized aviation medicine. Because the state of registry, the state of the operator, the state of design and the state of manufacture shall each be entitled to appoint an accredited representative to participate in the investigation we take account, additionally, the critical responses by these shareholders in two of the cases [10,11]. Also, safety recommendations to mitigate future similar kinds of accidents were analyzed regarding the two more recent accidents [12, 13]. |
|
4. It would be helpful to offer a critical response or view on the other two accidents, in addition to [10], [11]; even from broader literature sources (not necessarily government, e.g. ISASI or similar).
|
Our response Regarding Silk Air flight we have added the following reference: In two accidents, the accident investigation authorities that conducted the investigations and the aviation authority that commented on the investigation had different interpretations as to the cause of the accident (Table 3) [Flight Safety Foundation, 2001, 2018; 10; 11]. New references: Flight Safety Foundation. Accident Prevention. March 2001. Official report provides no conclusions about the cause of SilkAir B-737’s fatal plinge from 35,000 feet. https://flightsafety.org/ap/ap_mar01.pdf Flight Safety Foundation. Aviation Medicine. February. 2018. Coping the cockpit. https://flightsafety.org/asw-article/coping-in-the-cockpit/
|
|
Discussion 5. A sudden introduction of COVID-19 in the discussion. It would be good if there is a preliminary reference or mention earlier in the paper to be used here in the discussion.
|
Our response We have added a reference to the introduction – see kindly below:
Also, the impact of COVID-19 pandemic needs to be considered [15]. |
|
6. Well done on creating a checklist for Psychological autopsy (PA). Even better if you have clearly outlined the same and dedicated more explicit discussion/ presentation to the same.
|
Our response: We have added a sentence and reference to deepen the presentation of PA. Isometsä (2001) concluded in his review that PA is “one of the most valuable tools of research on completed suicide”. It can be concluded that it is synthesizing information from multiple resources. It can be applied to suicides that have occurred in other occupations such as in the building industry and medicine (Heller et al., 2007; Serebrenic et al., 2021).
Reference Isometsä ET. Psychological autopsy studies—a review. Eur Psychiatry. 2001 Nov;16(7):379-85. doi: 10.1016/s0924-9338(01)00594-6.
|
|
7. Why not introduce MH 370 in the beginning with the other accidents and then use it in the discussion?
|
Our response We have added a brief history of each accident mentioned in this study. Introduction Additionally, we mention in the discussion two accidents which potentially could be due to aircraft assisted suicides. The first is the Malaysian Airlines flight MH370 which on 8 March 2014 disappeared soon after a routine handover to Vietnam air traffic control. Despite an extensive search, the aircraft has not been located [REF]. The second is the China Eastern Airlines flight 5735 which on 21 March 2022 suddenly descended steeply mid-flight and struck the ground at high speed [REF]. The aircraft accident investigation is still ongoing. Materials and methods Four commercial aircraft accidents were included in this study. The first analyzed accident occurred in Indonesia on 19 December 1997 [10]. SilkAir flight MI 185 was an operating flight traveling from Jakarta Soekarno-Hatta International Airport to Singapore Changi Airport. The airplane descended from its cruising altitude of 35,000 feet and impacted the ground. The FDR stopped recording several times before the descent, and there were no mayday calls from the pilots. The second analyzed accident occurred on 31 October 1999, when EgyptAir flight 990 crashed into the Atlantic Ocean South of Massachusetts [11]. About 29 minutes after take-off the FDR showed that the first officer disconnected the autopilot. The FDR recorded an abrupt nose-down elevator movement, and the aircraft began to descend. The CVR recorded that the first officer said several times “I rely on God”. After that, the FDR recorded additional, slightly larger inboard aileron movements. A master warning alarm began to sound, and the captain who was returning from the toilet was wondering what was happening. The aircraft crashed about two minutes later. The third analyzed accident occurred on 29 November 2013 with Mozambique Airlines flight 470 [12]. After 1hour 50 minutes into the flight the First Officer stated he had to go to the toilet. Then the captain handled the auto flight system leading to a sustained descent and collision with the terrain. Just before the collision sounds were heard of someone trying to open the cockpit door. The fourth analyzed accident occurred on 24 March 2015 with Germanwings flight 4U9525 [13]. In the cruise phase of the flight, the first officer waited until he was alone in the cockpit. Then he modified the autopilot settings to order the aeroplane to descend and kept the cockpit door locked. The first officer did not respond to the calls from air traffic controllers, and the aircraft impacted the terrain in the French Alps. |
|
8. It would be good if there were some examples from other industries or cases where the psychological autopsy has already been used in other industries to indicate how useful it would be to introduce it into reviews of accidents in aviation.
|
Our response We have added examples
It can be applied to suicides that have occurred in other occupations such as in the building industry and health care (Heller et al., 2007; Serebrenic et al., 2021).
References
Heller TS, Hawgood JL, Leo DD. Correlates of suicide in building industry workers. Arch Suicide Res. 2007;11(1):105-17. doi: 10.1080/13811110600992977. Serebrenic F, Carmona MJC, Cunha PJ, Malbergier A. Postmortem qualitative analysis of psychological, occupational, and environmental factors associated with lethal anesthetic and/or opioid abuse among anesthesiologists: case series. Braz J Anesthesiol. 2021 Jul-Aug;71(4):317-325. doi: 10.1016/j.bjane.2021.05.002. |
|
9. Consider adding more details about the serotonin syndrome for a more generalist reader of the Safety journal in addition to the reference already offered.
|
Our response: We added the following text and references. The prevalence of this symptom is very low (in a large cohort study less than 0.2% of patients received serotonergic agents), and the symptom varies from mild to severe [Nguen et al., 2017; Simon et al., 2022]. Also, interactions of psychiatric drugs, as well between each other or with other drugs may cause behavioral changes which need to be taken into consideration(Butler, 2018).
References
Butler MG. Pharmacogenetics and Psychiatric Care: A Review and Commentary. J Ment Health Clin Psychol. 2018;2(2):17-24.
Nguyen CT, Xie L, Alley S, McCarron RM, Baser O, Wang Z. Epidemiology and Economic Burden of Serotonin Syndrome With Concomitant Use of Serotonergic Agents: A Retrospective Study Utilizing Two Large US Claims Databases. Prim Care Companion CNS Disord. 2017 Dec 28;19(6):17m02200. doi: 10.4088/PCC.17m02200 Simon LV, Keenaghan M. Serotonin Syndrome. [Updated 2022 Jul 19]. In: StatPearls [Internet]. Treasure Island (FL): StatPearls Publishing; 2022 Jan-. Available from: https://www.ncbi.nlm.nih.gov/books/NBK482377/
|
|
10. Comments at the end of the section state the utility of the findings from one of the accidents. Is it better if the authors offered more confirmation of this from the literature? As it stands now, it is unclear whether it’s the conclusion based on the feeling or arguments.
|
We have added sentence to the end of this paragraph.
As an example of this positive impact is EASA recent report of Mental Health for Aviation SAFEty (MESAFE) [EASA, 2022]. This report covers among other issues mental health assessment methods.
Reference European Aviation Safety Authority (EASA), 2022. Mental Health for Aviation SAFEty (MESAFE). https://www.easa.europa.eu/en/downloads/137506/en(Accessed 12 February 2023).
|
|
Conclusion 11. The conclusion states that imaging from the cockpit may help the investigation. It would be good to see from the literature earlier in the text if there is a case that this happens in some other industries. This practice might come from control rooms or laboratories, where these images help backtrack or find specific causes of incidents.
|
Our response. We have added text and reference: In summary, to prevent what may arguably be preventable pilot suicide accidents involving commercial aircraft, it is necessary to identify the causes of these accidents to be able to provide meaningful safety recommendations. As shown in the accident investigation of Germanwings flight 4U9525, a detailed psychological autopsy on pilots can and likely will assist in investigations, as well as generate recommendations that will substantially contribute to mitigate accidents due to pilot suicide. Airborne image recording may provide additional information about events before the crash which may be helpful in assisting in an accident investigation. Use of this registration should be expedited. This recording would have given additional valuable information especially in the accidents of Egypt Air flight 990 and Silk Air flight 185. Unfortunately, in an aviation medicine clinical setting over-reliance on the known risk-factors or use scales may create false reassurance but comprehensive psychosocial assessments when there are risk factors could be useful (Chan et al., 2016). There also is a need to carry out research on the detailed neuropsychiatric history targeted to those pilots who have had suicidal ideation (Rice and Sher, 2016).
References Chan MK, Bhatti H, Meader N, Stockton S, Evans J, O’Connor RC, Kapur N, Kendall T. Predicting suicide following self-harm: systematic review of risk factors and risk scales. Br J Psychiatry. 2016 Oct;209(4):277-283. doi: 10.1192/bjp.bp.115.170050.
Rice TR, Sher L. Preventing plane-assisted suicides through the lessons of research on homicide and suicide-homicide. Acta Neuropsychiatr. 2016 Aug;28(4):195-8. doi: 10.1017/neu.2015.67.
|
|
12. Apply a more formal and less conversational writing style throughout the paper (e.g. of course; ). Please check the attached PDF for further examples of this.
|
Separate comments: Page 2. See the red text. We clarified the paragraph. Because the state of registry, the state of the operator, the state of design and the state of manufacture shall each be entitled to appoint an accredited representative to participate in the investigation we take account, additionally, the critical responses by these shareholders in two of the cases [10,11]. Page 2. HFACS analytics was not used in these accident investigations. Page 3. We have provided the table as it is below but unfortunately the lay out was somehow changed. We will communicate this issue with the editor. We agree that table 3 shows that there was a lack of medical data accidents occurred in Massachusetts and Indonesia. Our response We have added sentence: Unfortunately, there was only limited amount of medical investigation data regarding both accidents. Page 4. The comment regarding criminal act is interesting but we have decided not to handle this topic in this study. Page 4. Comment regarding the table (Accidents occurred in Indonesia and Massachusetts) layout. We found that the table layout was somehow changed and we will communicate this with the journal editor. We do not think it helpful to add more data in this table as it would become cumbersome. Page 4 results table “Documents..”. Pulled has been substituted with pushed. We had footnote that somehow was lost in generated the submission. The footnote is the following: *Statement that no significant medical history was related to crew. ** Close friend interviewed. *** Information regarding medical certification and some social information under the title “Human factors”.
Page 5 Table “Safety…” the layout of the table was somehow changed and we will communicate this with the editor.
Page 6. Verifying an aircraft accident ïƒ verifying
Page 6. We prefer to use suicide (not murder suicide)
Page 6. Regarding ICAO and Annex 6 we were unable to find out whether all ICAO countries have been participating in this process.
Page 7. We think that there are so little toxicological data available that a table would not be helpful for the reader.
|
|
13. We have observed a good selection of references, although dated at times. It would be better if you used some more Safety Journal references, if available. Consider adding.
|
Our response Based on earlier comments we have already added nearly 20 new refences.
|
|
REVIEWER B
|
|
|
This study presents an interesting topic that requires further inquiry. Nonetheless, there are some concerns that need to be addressed prior to consideration of publication.
|
We thank you. |
|
The abstract lacks the purpose of the study.
|
Our response
We have added the aim of the study in the abstract.
The aim was to identify how the various aircraft accident investigators arrived at the conclusion that the accidents were due to suicidal acts.
|
|
Introduction
Not all readers will have an aviation background. Thus, the introduction section (readers) would significantly benefit from an explanation of the aircraft accident investigation processes and purposes, ideally using the International Civil Aviation Organization (ICAO) standards and recommended practices (i.e. ICAO Annex 13; ICAO Doc. 9756 Parts I / II / III). For example, according to the ICAO Annex 13, […] “The State conducting the investigation shall send a copy of the draft Final Report to the following States inviting their significant and substantiated comments on the report as soon as possible: a) the State that instituted the investigation; b) the State of Registry; c) the State of the Operator; d) the State of Design; e) the State of Manufacture; and f) any State that participated in the investigation as per Chapter 5”.
|
Our response
We have added the following to the introduction (second paragraph)
According to the ICAO Annex 13, [ICAO, 2020] “The State conducting the investigation shall send a copy of the draft Final Report to the following States inviting their significant and substantiated comments on the report as soon as possible: a) the State that instituted the investigation; b) the State of Registry; c) the State of the Operator; d) the State of Design; e) the State of Manufacture; and f) any State that participated in the investigation as per Chapter 5.
Reference
International Civil Aviation Organization. Annex 13, July 2020. Twelfth Edition. Montreal, Canada.
|
|
Readers would also benefit from a brief history of each aircraft accident, including information about the air carrier operator (i.e. from Indonesia), and why suicidal acts could have been a factor in each mishap.
|
Our response We have added a brief history of each accident mentioned in this study. Introduction Additionally, we mention in the discussion two accidents which potentially could be due to aircraft assisted suicides. The first is the Malaysian Airlines flight MH370 which on 8 March 2014 disappeared soon after a routine handover to Vietnam air traffic control. Despite an extensive search, the aircraft has not been located [REF]. The second is the China Eastern Airlines flight 5735 which on 21 March 2022 suddenly descended steeply mid-flight and struck the ground at high speed [REF]. The aircraft accident investigation is still ongoing. Materials and methods Four commercial aircraft accidents were included in this study. The first analyzed accident occurred in Indonesia on 19 December 1997 [10]. SilkAir flight MI 185 was an operating flight traveling from Jakarta Soekarno-Hatta International Airport to Singapore Changi Airport. The airplane descended from its cruising altitude of 35,000 feet and impacted the ground. The FDR stopped recording several times before the descent, and there were no mayday calls from the pilots. The second analyzed accident occurred on 31 October 1999, when EgyptAir flight 990 crashed into the Atlantic Ocean South of Massachusetts [11]. About 29 minutes after take-off the FDR showed that the first officer disconnected the autopilot. The FDR recorded an abrupt nose-down elevator movement, and the aircraft began to descend. The CVR recorded that the first officer said several times “I rely on God”. After that, the FDR recorded additional, slightly larger inboard aileron movements. A master warning alarm began to sound, and the captain who was returning from the toilet was wondering what was happening. The aircraft crashed about two minutes later. The third analyzed accident occurred on 29 November 2013 with Mozambique Airlines flight 470 [12]. After 1hour 50 minutes into the flight the First Officer stated he had to go to the toilet. Then the captain handled the auto flight system leading to a sustained descent and collision with the terrain. Just before the collision sounds were heard of someone trying to open the cockpit door. The fourth analyzed accident occurred on 24 March 2015 with Germanwings flight 4U9525 [13]. In the cruise phase of the flight, the first officer waited until he was alone in the cockpit. Then he modified the autopilot settings to order the aeroplane to descend and kept the cockpit door locked. The first officer did not respond to the calls from air traffic controllers, and the aircraft impacted the terrain in the French Alps. |
|
Readers would also benefit from information on the challenges by aircraft accident investigators (or other professionals) to determine that suicide was a causal factor of an (i.e. aircraft / car) accident. Evidence from previous studies is recommended.
|
Our response We have added the text in the first paragraph of introduction.
Prior to the Germanwings aircraft assisted suicide in 2015, there have been only a few research articles that have analyzed aircraft assisted suicides and these analyses focused on general rather than on commercial aviation (Bills et al., 2005; Lewis et al., 2007; Schwark et al., 2008; Cullen et al., 2007). In general aviation, there is generally no cockpit voice recorder (CVR) and flight data recorder (FDR) data. In these reports, analysis of aircraft accident suicides in general aviation is based on autopsy and pilot medical history which in many cases is incomplete and brief, which may contribute to inconclusive or best assumption from the evidence of findings of suicide. References Bills CB, Crabowski JG, Li G. Suicide by aircraft: a comparative analysis. Aviat Space Environ Med 2005; 76:715–9. LewisRJ, JohnsonRD, WhinneryJE, ForsterEM. Aircraft-assisted pilot suicides in the United States, 1993-2002. Arch Suicide Res 2007; 11:149–61 Schwark T, Severin K, Grellner W. “I am flying to the stars”— suicide by aircraft in Germany. Forensic Sci Int 2008; 179:e75–8. Cullen SA, Drysdale HC, Mayes RW. Role of medical factors in 1000 fatal accidents: case note study. BMJ 1997; 314:1592.
|
|
Few citations are needed throughout the manuscript. For example, […] “Furthermore, many cases involve private (single acting) pilots or those of very small operators where investigations may be more limited, as opposed to the rarer intentional crashes brought about with large passenger aircraft”. Citation(s)?
|
Our response
We have added the reference and modified the sentence
Furthermore, many cases involve private (single acting) pilots or those of very small operators where investigations may be more limited as shown in analysis of accidents related to aircraft assisted suicides in USA [Vuorio et al.., 2014], as opposed to the rarer intentional crashes brought about with large passenger aircraft.
Reference
Vuorio A, Laukkala T, Navathe P, Budowle B, Eyre A, Sajantila A. Aircraft-assisted pilot suicides: lessons to be learned. Aviat. Space Environ. Med. 2014, 85: 841-6.
|
|
Authors explained “psychological autopsy” in the discussion section of the manuscript. This explanation should be in the introduction section of the paper.
|
Our response
We have added an explanation of psychological autopsy in the introduction
Collecting socio-economic background can be comparable to that of a psychological autopsy (PA) which is a forensic assessment to evaluate the deceased’s mental ‘condition’ (as ‘state’ cannot be assessed retrospectively unless the pilot was seen by a medical practitioner shortly before the crash) and likely intent. This may include rehearsals for the act such as through flight simulator scenarios being tested and the social and financial circumstances of the pilot being investigated to see whether these may be linked to personal life challenges and emotional distress. Also, the impact of COVID-19 pandemic needs to be considered [15].
|
|
Table 1 may not be needed. Its content could be presented in text.
|
Our response
Table 1 has been deleted.
|
|
Authors suggested that […] “based on these analyses and experience, recommendations to mitigate future aircraft assisted suicides are considered to investigate accidents in which there are suspicions of possible aircraft assisted suicide”. Further explanation on “experience” is needed!
|
Our response
We have added the refence to the end of this sentence and modified it.
There also is a need to carry out research on the detailed neuropsychiatric history targeted to those pilots who have had suicidal ideation (Rice and Sher, 2016). |
|
Materials and Methods
The methodology utilized by the authors resembles a qualitative case study that would enable them to conduct an in-depth exploration of intricate phenomena within some specific context. Further information on the procedures to collect and especially analyze data and information is recommended (i.e. triangulation? / content analysis?). What were the roles of the researchers during the data analysis?
|
Our response
We have added clarifying sentence
We used content analysis and one of the authors (AV) analyzed in detail the accident reports and other authors commented the analysis. |
|
There are two “Table 3”.
|
Corrected |
|
Table 3. Documented medical and pathological information collected in accident investigations [10-13]”. No need for this Table – information could be provided in text.
|
Our response
We prefer to keep this table to convey how little medical information was available.
|
|
“Table 3. Accidents Occurred in Indonesia and Massachusetts: Official Accident Report vs. Response to Accident Investigation Draft Report [10-13]”. This Table needs a different format for a better understanding (i.e. different page orientation?).
|
Our response
We agree and will communicate this with the editor. The layout was somehow modified during submission.
|
|
Table 3 – Accidents Occurred in Indonesia and Massachusetts: Official Accident Report vs. Response to Accident Investigation Draft Report [10-13]”. Authors should provide in text more information (a brief explanation) on its contents.
|
Our response
We have added the sentence
According to ICAO Annex 13 protocol the State of Registry and State of Manufacturer also have the possibility to comment on the investigation. |
|
Table 4 – Not mentioned nor explained in text.
|
Our response
Added in text.
|
|
Other Recommendations
(BEA, 2015) – Different type of citation
|
Corrected
|
|
Some abbreviations need explanation in text (i.e. ICAO; BEA).
|
Corrected
|
|
A grammar and syntax review in the entire manuscript is highly recommended. For example, “the safety recommendations because of the accident in Namibia were primarily oriented towards solutions related to the technical operation and the tracking of the aircraft [12]” / “the safety recommendations because of the accident in France were related to the medical monitoring of pilots and the expansion of emotional support for pilots [13]” / “as an example, an accident under investigation is flight MU5735 occurred 21 March 2021” may need rewording.
|
We have throughout the manuscript improved the grammar.
|
|
Discussions
Authors stated that “based on our experience, we created a checklist for the PA to be carried out in aviation medicine related mental health cases”. This is NOT the purpose of their study!
|
Our response
We changed the first sentence of this paragraph to avoid this kind of idea.
PA is useful to use in aviation medicine related mental health cases. |
|
Conclusion
Authors should present the limitations of their study. It is also suggested authors provide recommendation for future studies.
|
Our response:
We have added limitations and also suggestion for future research.
There are some limitations in this study. Although all analyzed accident investigations followed ICAO Annex 13 guidelines, there is variability in their accident investigations and reporting. This variation makes the comparisons somewhat difficult. In addition, accident investigation reports have evolved during the years 1997 to 2015. Currently the purpose is not to find individual causes of the accident. Instead, the investigation report includes a description of the accident, the factors that led to the accident, and the consequences of the accident, as well as safety recommendations addressed to the relevant authorities and other actors for such measures as are necessary to increase public safety, prevent new accidents and dangerous situations, combat damage, and enhance the operations of rescue and other authorities.
Recommendation We also need research carrying out a detailed neuropsychiatric history targeted to those pilots who have had suicidal ideation (Rice and Sher, 2016). Congratulations on the interesting research study. This project is a worthwhile endeavor. Nonetheless, I recommend authors should address the aforementioned concerns before this manuscript is accepted for publication. Thank you! |
|
Reviewer
|
|
|
I have reviewed the article “Commercial Aircraft Assisted Suicide Accident Investigations Re- visited –Agreeing to disagree?”. Even if the topic is one of a maximum value and impact for the aviation community and your endeavor is very much appreciated, the conclusions of your analysis are either too general (“it is necessary to identify the causes of these accidents to be able to provide meaningful safety recommendations” – is it not the major goal of any air crash investigation committee to identify the causes of the accident?) or they are already included in most of the air crash investigations (“A detailed psychological autopsy on pilots can and likely will assist in investigations” – this is quite often, if not always, performed by the air crash investigation committees).
|
Our response We have modified the sentence Unlike more specific and arguably more clearcut physiological determinants or the course of events, causes and consequences of an air crash, such as cardiac or neurological symptoms, mental health and well-being factors, are less straightforward to determine. |
|
In what concerns your third conclusion – “Airborne image recording may provide additional information about events before the crash which may be helpful in assisting in an accident investigation.” – you already mentioned at page 6, the last but one paragraph, that “Currently, there is an attempt by the ICAO to mandate airborne image recorders”. So, this conclusion also brings nothing new as there is already an ongoing procedure to mandate the usage of these recorders.
|
Our response
We have added text
Use of this registration should be expedited. This recording would have given additional valuable information especially in the accidents of Egypt Air flight 990 and Silk Air flight 185. |
|
I think that your decision to study this topic in general and these four reports in particular, was an inspired one, but perhaps a new, in depth analysis will help you reach more original conclusions that can be better tailored to the need of spotting out the pilots with suicidal behavior. Particular patterns that may be useful to track down in advance the pilots at risk to commit suicide, could perhaps, be identified within these reports.
|
Our response To the best of our knowledge this kind of analysis has not been carried previously and consider it to be an in depth analysis, although the number of cases is limited.
|
|
Under the present form, I am afraid that I cannot endorse the publication of your article.
|
Our response We have improved the study with the aid of two other referees and hopefully, the reviewers will find that it has been improved.
|
|
|
|
Reviewer 3 Report
See Attached Notes

Author Response

(The authors gave the same response as above.)

Reviewer 4 Report
This study presents an interesting topic that requires further inquiry. Nonetheless, there are some concerns that need to be addressed prior to consideration of publication.
Abstract
The abstract lacks the purpose of the study.
Introduction
Not all readers will have an aviation background. Thus, the introduction section (readers) would significantly benefit from an explanation of the aircraft accident investigation processes and purposes, ideally using the International Civil Aviation Organization (ICAO) standards and recommended practices (i.e. ICAO Annex 13; ICAO Doc. 9756 Parts I / II / III). For example, according to the ICAO Annex 13, […] “The State conducting the investigation shall send a copy of the draft Final Report to the following States inviting their significant and substantiated comments on the report as soon as possible:
a) the State that instituted the investigation;
b) the State of Registry;
c) the State of the Operator;
d) the State of Design;
e) the State of Manufacture; and
f) any State that participated in the investigation as per Chapter 5”.
Readers would also benefit from a brief history of each aircraft accident, including information about the air carrier operator (i.e. from Indonesia), and why suicidal acts could have been a factor in each mishap.
Readers would also benefit from information on the challenges by aircraft accident investigators (or other professionals) to determine that suicide was a causal factor of an (i.e. aircraft / car) accident. Evidence from previous studies is recommended.
Few citations are needed throughout the manuscript. For example, […] “Furthermore, many cases involve private (single acting) pilots or those of very small operators where investigations may be more limited, as opposed to the rarer intentional crashes brought about with large passenger aircraft”. Citation(s)?
Authors explained “psychological autopsy” in the discussion section of the manuscript. This explanation should be in the introduction section of the paper.
Table 1 may not be needed. Its content could be presented in text.
Authors suggested that […] “based on these analyses and experience, recommendations to mitigate future aircraft assisted suicides are considered to investigate accidents in which there are suspicions of possible aircraft assisted suicide”. Further explanation on “experience” is needed!
Materials and Methods
The methodology utilized by the authors resembles a qualitative case study that would enable them to conduct an in-depth exploration of intricate phenomena within some specific context. Further information on the procedures to collect and especially analyze data and information is recommended (i.e. triangulation? / content analysis?). What were the roles of the researchers during the data analysis?
Results
There are two “Table 3”.
“Table 3. Documented medical and pathological information collected in accident investigations
[10-13]”. No need for this Table – information could be provided in text.
“Table 3. Accidents Occurred in Indonesia and Massachusetts: Official Accident Report vs. Response to Accident Investigation Draft Report [10-13]”. This Table needs a different format for a better understanding (i.e. different page orientation?).
Table 3 – Accidents Occurred in Indonesia and Massachusetts: Official Accident Report vs. Response to Accident Investigation Draft Report [10-13]”. Authors should provide in text more information (a brief explanation) on its contents.
Table 4 – Not mentioned nor explained in text.
Other Recommendations
(BEA, 2015) – Different type of citation
Some abbreviations need explanation in text (i.e. ICAO; BEA).
A grammar and syntax review in the entire manuscript is highly recommended. For example, “the safety recommendations because of the accident in Namibia were primarily oriented towards solutions related to the technical operation and the tracking of the aircraft [12]” / “the safety recommendations because of the accident in France were related to the medical monitoring of pilots and the expansion of emotional support for pilots [13]” / “as an example, an accident under investigation is flight MU5735 occurred 21 March 2021” may need rewording.
Discussions
Authors stated that “based on our experience, we created a checklist for the PA to be carried out in aviation medicine related mental health cases”. This is NOT the purpose of their study!
Conclusion
Authors should present the limitations of their study. It is also suggested authors provide recommendation for future studies.
Congratulations on the interesting research study. This project is a worthwhile endeavor. Nonetheless, I recommend authors should address the aforementioned concerns before this manuscript is accepted for publication.
Author Response
|
REVIEWER B
|
|
|
This study presents an interesting topic that requires further inquiry. Nonetheless, there are some concerns that need to be addressed prior to consideration of publication.
|
We thank you. |
|
The abstract lacks the purpose of the study.
|
Our response
We have added the aim of the study in the abstract.
The aim was to identify how the various aircraft accident investigators arrived at the conclusion that the accidents were due to suicidal acts.
|
|
Introduction
Not all readers will have an aviation background. Thus, the introduction section (readers) would significantly benefit from an explanation of the aircraft accident investigation processes and purposes, ideally using the International Civil Aviation Organization (ICAO) standards and recommended practices (i.e. ICAO Annex 13; ICAO Doc. 9756 Parts I / II / III). For example, according to the ICAO Annex 13, […] “The State conducting the investigation shall send a copy of the draft Final Report to the following States inviting their significant and substantiated comments on the report as soon as possible: a) the State that instituted the investigation; b) the State of Registry; c) the State of the Operator; d) the State of Design; e) the State of Manufacture; and f) any State that participated in the investigation as per Chapter 5”.
|
Our response
We have added the following to the introduction (second paragraph)
According to the ICAO Annex 13, [ICAO, 2020] “The State conducting the investigation shall send a copy of the draft Final Report to the following States inviting their significant and substantiated comments on the report as soon as possible: a) the State that instituted the investigation; b) the State of Registry; c) the State of the Operator; d) the State of Design; e) the State of Manufacture; and f) any State that participated in the investigation as per Chapter 5.
Reference
International Civil Aviation Organization. Annex 13, July 2020. Twelfth Edition. Montreal, Canada.
|
|
Readers would also benefit from a brief history of each aircraft accident, including information about the air carrier operator (i.e. from Indonesia), and why suicidal acts could have been a factor in each mishap.
|
Our response We have added a brief history of each accident mentioned in this study. Introduction Additionally, we mention in the discussion two accidents which potentially could be due to aircraft assisted suicides. The first is the Malaysian Airlines flight MH370 which on 8 March 2014 disappeared soon after a routine handover to Vietnam air traffic control. Despite an extensive search, the aircraft has not been located [REF]. The second is the China Eastern Airlines flight 5735 which on 21 March 2022 suddenly descended steeply mid-flight and struck the ground at high speed [REF]. The aircraft accident investigation is still ongoing. Materials and methods Four commercial aircraft accidents were included in this study. The first analyzed accident occurred in Indonesia on 19 December 1997 [10]. SilkAir flight MI 185 was an operating flight traveling from Jakarta Soekarno-Hatta International Airport to Singapore Changi Airport. The airplane descended from its cruising altitude of 35,000 feet and impacted the ground. The FDR stopped recording several times before the descent, and there were no mayday calls from the pilots. The second analyzed accident occurred on 31 October 1999, when EgyptAir flight 990 crashed into the Atlantic Ocean South of Massachusetts [11]. About 29 minutes after take-off the FDR showed that the first officer disconnected the autopilot. The FDR recorded an abrupt nose-down elevator movement, and the aircraft began to descend. The CVR recorded that the first officer said several times “I rely on God”. After that, the FDR recorded additional, slightly larger inboard aileron movements. A master warning alarm began to sound, and the captain who was returning from the toilet was wondering what was happening. The aircraft crashed about two minutes later. The third analyzed accident occurred on 29 November 2013 with Mozambique Airlines flight 470 [12]. After 1hour 50 minutes into the flight the First Officer stated he had to go to the toilet. Then the captain handled the auto flight system leading to a sustained descent and collision with the terrain. Just before the collision sounds were heard of someone trying to open the cockpit door. The fourth analyzed accident occurred on 24 March 2015 with Germanwings flight 4U9525 [13]. In the cruise phase of the flight, the first officer waited until he was alone in the cockpit. Then he modified the autopilot settings to order the aeroplane to descend and kept the cockpit door locked. The first officer did not respond to the calls from air traffic controllers, and the aircraft impacted the terrain in the French Alps. |
|
Readers would also benefit from information on the challenges by aircraft accident investigators (or other professionals) to determine that suicide was a causal factor of an (i.e. aircraft / car) accident. Evidence from previous studies is recommended.
|
Our response We have added the text in the first paragraph of introduction.
Prior to the Germanwings aircraft assisted suicide in 2015, there have been only a few research articles that have analyzed aircraft assisted suicides and these analyses focused on general rather than on commercial aviation (Bills et al., 2005; Lewis et al., 2007; Schwark et al., 2008; Cullen et al., 2007). In general aviation, there is generally no cockpit voice recorder (CVR) and flight data recorder (FDR) data. In these reports, analysis of aircraft accident suicides in general aviation is based on autopsy and pilot medical history which in many cases is incomplete and brief, which may contribute to inconclusive or best assumption from the evidence of findings of suicide. References Bills CB, Crabowski JG, Li G. Suicide by aircraft: a comparative analysis. Aviat Space Environ Med 2005; 76:715–9. LewisRJ, JohnsonRD, WhinneryJE, ForsterEM. Aircraft-assisted pilot suicides in the United States, 1993-2002. Arch Suicide Res 2007; 11:149–61 Schwark T, Severin K, Grellner W. “I am flying to the stars”— suicide by aircraft in Germany. Forensic Sci Int 2008; 179:e75–8. Cullen SA, Drysdale HC, Mayes RW. Role of medical factors in 1000 fatal accidents: case note study. BMJ 1997; 314:1592.
|
|
Few citations are needed throughout the manuscript. For example, […] “Furthermore, many cases involve private (single acting) pilots or those of very small operators where investigations may be more limited, as opposed to the rarer intentional crashes brought about with large passenger aircraft”. Citation(s)?
|
Our response
We have added the reference and modified the sentence
Furthermore, many cases involve private (single acting) pilots or those of very small operators where investigations may be more limited as shown in analysis of accidents related to aircraft assisted suicides in USA [Vuorio et al.., 2014], as opposed to the rarer intentional crashes brought about with large passenger aircraft.
Reference
Vuorio A, Laukkala T, Navathe P, Budowle B, Eyre A, Sajantila A. Aircraft-assisted pilot suicides: lessons to be learned. Aviat. Space Environ. Med. 2014, 85: 841-6.
|
|
Authors explained “psychological autopsy” in the discussion section of the manuscript. This explanation should be in the introduction section of the paper.
|
Our response
We have added an explanation of psychological autopsy in the introduction
Collecting socio-economic background can be comparable to that of a psychological autopsy (PA) which is a forensic assessment to evaluate the deceased’s mental ‘condition’ (as ‘state’ cannot be assessed retrospectively unless the pilot was seen by a medical practitioner shortly before the crash) and likely intent. This may include rehearsals for the act such as through flight simulator scenarios being tested and the social and financial circumstances of the pilot being investigated to see whether these may be linked to personal life challenges and emotional distress. Also, the impact of COVID-19 pandemic needs to be considered [15].
|
|
Table 1 may not be needed. Its content could be presented in text.
|
Our response
Table 1 has been deleted.
|
|
Authors suggested that […] “based on these analyses and experience, recommendations to mitigate future aircraft assisted suicides are considered to investigate accidents in which there are suspicions of possible aircraft assisted suicide”. Further explanation on “experience” is needed!
|
Our response
We have added the refence to the end of this sentence and modified it.
There also is a need to carry out research on the detailed neuropsychiatric history targeted to those pilots who have had suicidal ideation (Rice and Sher, 2016). |
|
Materials and Methods
The methodology utilized by the authors resembles a qualitative case study that would enable them to conduct an in-depth exploration of intricate phenomena within some specific context. Further information on the procedures to collect and especially analyze data and information is recommended (i.e. triangulation? / content analysis?). What were the roles of the researchers during the data analysis?
|
Our response
We have added clarifying sentence
We used content analysis and one of the authors (AV) analyzed in detail the accident reports and other authors commented the analysis. |
|
There are two “Table 3”.
|
Corrected |
|
Table 3. Documented medical and pathological information collected in accident investigations [10-13]”. No need for this Table – information could be provided in text.
|
Our response
We prefer to keep this table to convey how little medical information was available.
|
|
“Table 3. Accidents Occurred in Indonesia and Massachusetts: Official Accident Report vs. Response to Accident Investigation Draft Report [10-13]”. This Table needs a different format for a better understanding (i.e. different page orientation?).
|
Our response
We agree and will communicate this with the editor. The layout was somehow modified during submission.
|
|
Table 3 – Accidents Occurred in Indonesia and Massachusetts: Official Accident Report vs. Response to Accident Investigation Draft Report [10-13]”. Authors should provide in text more information (a brief explanation) on its contents.
|
Our response
We have added the sentence
According to ICAO Annex 13 protocol the State of Registry and State of Manufacturer also have the possibility to comment on the investigation. |
|
Table 4 – Not mentioned nor explained in text.
|
Our response
Added in text.
|
|
Other Recommendations
(BEA, 2015) – Different type of citation
|
Corrected
|
|
Some abbreviations need explanation in text (i.e. ICAO; BEA).
|
Corrected
|
|
A grammar and syntax review in the entire manuscript is highly recommended. For example, “the safety recommendations because of the accident in Namibia were primarily oriented towards solutions related to the technical operation and the tracking of the aircraft [12]” / “the safety recommendations because of the accident in France were related to the medical monitoring of pilots and the expansion of emotional support for pilots [13]” / “as an example, an accident under investigation is flight MU5735 occurred 21 March 2021” may need rewording.
|
We have throughout the manuscript improved the grammar.
|
|
Discussions
Authors stated that “based on our experience, we created a checklist for the PA to be carried out in aviation medicine related mental health cases”. This is NOT the purpose of their study!
|
Our response
We changed the first sentence of this paragraph to avoid this kind of idea.
PA is useful to use in aviation medicine related mental health cases. |
|
Conclusion
Authors should present the limitations of their study. It is also suggested authors provide recommendation for future studies.
|
Our response:
We have added limitations and also suggestion for future research.
There are some limitations in this study. Although all analyzed accident investigations followed ICAO Annex 13 guidelines, there is variability in their accident investigations and reporting. This variation makes the comparisons somewhat difficult. In addition, accident investigation reports have evolved during the years 1997 to 2015. Currently the purpose is not to find individual causes of the accident. Instead, the investigation report includes a description of the accident, the factors that led to the accident, and the consequences of the accident, as well as safety recommendations addressed to the relevant authorities and other actors for such measures as are necessary to increase public safety, prevent new accidents and dangerous situations, combat damage, and enhance the operations of rescue and other authorities.
Recommendation We also need research carrying out a detailed neuropsychiatric history targeted to those pilots who have had suicidal ideation (Rice and Sher, 2016). Congratulations on the interesting research study. This project is a worthwhile endeavor. Nonetheless, I recommend authors should address the aforementioned concerns before this manuscript is accepted for publication. Thank you! |
Round 2
Reviewer 1 Report
The content has been improved. The article may be published in present form.
Author Response
Thank you for the final comment.
BW
Alpo Vuorio
Reviewer 4 Report
Dear Authors;
A few recommendations before your manuscript can be published.
1. Abstract - You defined FDR twice;
2. Limitations of the study - Should follow the "conclusion" section of the manuscript.
Congratulations on the interesting research manuscript.
Author Response
Thank you for the final comments. We have made required two corrections.
BW
Alpo Vuorio